# The Impact of Multiple Sclerosis Disease Status and Subtype on Hematological Profile

**DOI:** 10.3390/ijerph18063318

**Published:** 2021-03-23

**Authors:** Jacob M. Miller, Jeremy T. Beales, Matthew D. Montierth, Farren B. Briggs, Scott F. Frodsham, Mary Feller Davis

**Affiliations:** 1Department of Microbiology and Molecular Biology, Brigham Young University, Provo, UT 84602, USA; 43jmills@gmail.com (J.M.M.); beales.jeremy@gmail.com (J.T.B.); mattmontierth@gmail.com (M.D.M.); 2Department of Population and Quantitative Health Sciences, Case Western Reserve University, Cleveland, OH 44106, USA; farren.briggs@case.edu; 3Division of Nephrology and Hypertension, University of Utah, Salt Lake City, UT 84112, USA; scott.frod@gmail.com; 4Department of Biomedical Informatics, Vanderbilt University, Nashville, TN 37235, USA

**Keywords:** multiple sclerosis, random forest, electronic health records

## Abstract

Multiple sclerosis (MS) is an immune-mediated, demyelinating disease of the central nervous system. In this study, an MS cohort and healthy controls were stratified into Caucasian and African American groups. Patient hematological profiles—composed of complete blood count (CBC) and complete metabolic panel (CMP) test values—were analyzed to identify differences between MS cases and controls and between patients with different MS subtypes. Additionally, random forest models were used to determine the aggregate utility of common hematological tests in determining MS disease status and subtype. The most significant and relevant results were increased bilirubin and creatinine in MS cases. The random forest models achieved some success in differentiating between MS cases and controls (AUC values: 0.725 and 0.710, respectively) but were not successful in differentiating between subtypes. However, larger samples that adjust for possible confounding variables, such as treatment status, may reveal the value of these tests in differentiating between MS subtypes.

## 1. Introduction

Multiple sclerosis (MS) is a complex disease of the central nervous system in which the myelin sheaths of the neurons in the brain and spinal cord are damaged. As presentation of the disease varies widely between patients, several subtypes of MS have been defined based on patterns of its progression. Relapsing remitting multiple sclerosis (RRMS), the most common form of MS, is characterized by unpredictable attacks (with potentially permanent deficits) followed by periods of disease quiescence [1]. Over time, RRMS patients typically transition into secondary progressive multiple sclerosis (SPMS), which is characterized by steady disease progression. In contrast, a minority of MS cases are classified as primary progressive multiple sclerosis (PPMS), in which disability accrues from disease onset without relapses. PPMS patients constitute only about 10% of all MS patients [2].

Characterizing physiological differences present in MS and its subtypes has been of significant interest in the MS research community [3,4,5,6]. A number of case–control studies have reported cerebrospinal fluid (CSF) and hematological biomarkers associated with MS disease status [7,8,9,10]. Such biomarkers may also have utility as predictors of MS clinical features such as disease progression. Past work has uncovered CSF and hematological biomarkers that correlate with MS subtype [6,11,12,13]. In addition to fluid biomarkers associated with MS subtype, one study has found low-frequency genetic variants that influence MS subtype susceptibility [14]. While a number of subtype biomarkers have been reported in the literature, there are currently no predictive biomarkers for MS disease course in clinical use. In light of this scarcity of clinical predictors of disease course, and to further investigate physiological changes in MS patients in general, we performed a survey study of hematological profiles in MS patients and controls using commonly employed hematological panels. Additionally, we used random forest classifiers to determine the predictive potential of these blood panels in the context of MS case status and disease subtype. Random forest classifiers have been used to discover novel disease-biomarker relationships and to classify patients in a variety of settings [15,16,17], as well as in other clinically relevant applications [18,19].

In our analyses, we utilized two common hematological panels, the complete blood count (CBC) panel and comprehensive metabolic panel (CMP). Both panels are routine measurements used by clinicians to understand the overall health of a patient [20]. These tests are performed frequently on both ill and healthy patients. The CBC measures blood cell values such as hemoglobin, platelet count, and white blood cell count. The tests in the CMP quantify clinical chemistry values, such as blood serum levels of albumin, various ions, and several liver enzymes. Several values measured in these panels have been shown to be correlated with MS disease status, subtype, and disease progression [21,22,23]. In particular, studies have found an association between the neutrophil-to-lymphocyte ratio (NLR) and MS disease status, disability, and subtype [23,24,25,26,27]. Creatinine and bilirubin have also been implicated in similar studies [10,21,22,28]. In this study, we performed three analyses: first, to better understand the physiological differences present in MS patients in general, we investigated whether differences exist in CBC and CMP values between MS cases and controls. Al-Hussain et al. performed a similar analysis in 2017 using a small MS cohort and a subset of CBC and CMP tests [26]. We used a larger dataset to attempt to replicate their findings for this subset of tests and discover novel associations involving tests not included in their study. Second, due to the lack of clinical biomarkers of MS subtype and the widespread clinical use of the CBC and CMP, we investigated differences in hematological profiles between RRMS/SPMS and PPMS patients. Finally, to evaluate the overall utility of these blood panels in differentiating between MS cases and controls and between PPMS and RRMS/SPMS patients, we trained random forest classifiers using a subset of patients and tested the classifiers on the remaining patients.

## 2. Materials and Methods

### 2.1. Sample Population and Data Preprocessing

Laboratory values for MS patients and control patients were retrieved from de-identified health records in Vanderbilt University Medical Center’s Synthetic Derivative (SD). The SD also provided the patient race (observer recorded) and sex data used in the analyses. Patients with missing demographic values, a reported race other than African American or Caucasian, or multiple reported races were excluded from the analysis. MS subtype was determined for each patient using previously published extraction algorithms [29]. Only patients with one of the three major MS subtypes (PPMS, RRMS, or SPMS) were included in the final dataset, with RRMS and SPMS patients grouped together to stratify the cohort into relapsing and progressive groups. Demographic characteristics of the groups are noted in Table 1.

The two blood panels (CBC and CMP) collectively contain patient lab values for 35 biomarkers. Median values were calculated for each patient and used in all analyses. Neutrophil-to-lymphocyte ratio (NLR) measurements were calculated for each patient with a value for both neutrophil absolute count and lymphocyte absolute count on a given date, and the median ratio was used. Patient age at the time of the most recent measurement was used as a covariate in the analyses. Lastly, the data were stratified into Caucasian and African American groups for statistical analysis. For both groups, the group averages of all lab values were within the reference ranges. Not all patients had data for each biomarker; the number of patients ultimately used for each analysis can be found in Appendix A.

### 2.2. Statistical Methods

Logistic regression analysis was performed in R (R Core Team, Vienna, Austria, version 3.1.3). Each of the 36 lab tests was analyzed separately as an independent variable. For the case–control analysis, patient median value for the given lab test was used to predict MS disease status. For the subtype analysis, patient median value for the given lab test was used to predict MS subtype. Gender and patient age were included as covariates in all analyses. As seen results, some analyses were not performed in the African American group due to insufficient sample sizes.

To correct for multiple testing, Bonferroni correction (α = 0.05) was applied at the group level in each of the analyses (four total adjusted *p*-values were calculated). The adjusted *p*-value calculated for the Caucasian case–control analysis, the African American case–control analysis, and the Caucasian group in the subtype analysis was 0.0014. The adjusted *p*-value calculated for the African American group in the subtype analysis (with fewer tests) was 0.0017. Analysis results are reported below. Multiple regressions were performed with biologically related lab tests. Mean platelet volume and platelet count were analyzed in one regression. White blood cell count and absolute neutrophil count were analyzed in another regression.

To measure the utility of the CBC and CMP blood panels in classifying patients as MS cases or controls, as well as differentiating between MS subtypes, a random forest model was fitted to each of the previously described study populations. These analyses were restricted to include only individuals without missing values for each laboratory test, and tests with a significant number of missing values were excluded altogether. The analysis utilized the package “randomForest” in R (standard parameters were used). Receiver operating characteristic (ROC) curves and area under the curve (AUC) values were generated using the “ROCR” package to assess the performance of each random forest model in classifying subjects.

## 3. Results

### 3.1. Case–control Analysis

The results of the case–control analysis and subtype analysis are displayed in Table 2 and Table 3, respectively, for both the Caucasian (C) and African American (AA) groups. In the case–control analysis, hemoglobin levels (*p*-values: 1.84 × 10^−39^ (C) and 1.03 × 10^−5^ (AA)) and mean platelet volume (*p*-values: 1.61 × 10^−6^ (C) and 1.05 × 10^−4^ (AA)) were significant in both groups, with an association with MS. Packed cell volume (*p*-values: 2.80 × 10^−49^ (C) and 9.10 × 10^−9^ (AA)), red blood cell count (*p*-values: 1.38 × 10^−8^ (C) and 7.77 × 10^−6^ (AA)), and anion gap (*p*-values: 8.11 × 10^−37^ (C) and 1.17 × 10^−6^ (AA)) showed the same significant trend. Conversely, white blood cell count (*p*-values: 4.55 × 10^−19^ (C) and 1.67 × 10^−4^ (AA)) and albumin (*p*-values: 2.88 × 10^−16^ (C) and 1.05 × 10^−4^ (AA)) levels had significant protective effects in both groups. Bilirubin (*p*-values: 3.7 × 10^−104^ (C) and 1.50 × 10^−14^ (AA)) was the most significant biomarker and had the lowest odds ratio in both the Caucasian group (OR: 0.19, 95% CI: 0.16–0.22) and the African American group (OR: 0.19, 95% CI: 0.13–0.29).

Red cell distribution width (*p*-values: 1.19 × 10^−4^ (C) and 0.003 (AA)), neutrophil-to-lymphocyte ratio (*p*-values: 4.93 × 10^−16^ (C) and 0.042 (AA)), and creatinine (*p*-values: 2.32 × 10^−16^ (C) and 0.012 (AA)) reached significance in the Caucasian group and nominal significance in the African American group. As with WBC and bilirubin, an increase in either led to decreased risk of having MS in both groups. Creatinine had odds ratios of 0.48 (95% CI: 0.40–0.57) and 0.79 (95% CI: 0.64–0.92) in the two groups, considerably lower than any of the other biomarkers except bilirubin. Chloride (*p*-value: 1.66 × 10^−13^ (C)) and neutrophil absolute count (*p*-value: 5.36 × 10^−27^ (C)) reached significance in the Caucasian cohort but failed to reach significance in the African American cohort.

Multiple regression was performed for significant, related lab tests. In the CBC, both mean platelet volume (*p*-values: 1.5 × 10^−8^ (C) and 3.8 × 10^−6^ (AA)) and platelet count (*p*-values: 5.6 × 10^−5^ (C) and 0.005 (AA)) remained significant when regressed together. In a subsequent regression, neutrophil absolute count (*p*-value: 1.72 × 10^−11^) remained significant in the Caucasian cohort, while white blood cell count did not (*p*-value: 0.077). The reverse was seen in the African American cohort, with only white blood cell count remaining significant (*p*-value: 4.28 × 10^−5^).

### 3.2. Subtype Analysis

In the subtype analysis, no test was significant after Bonferroni correction, but several lab values were nominally significant in both groups. For the odds ratios, a larger odds ratio represents an increased risk of PPMS relative to RRMS/SPMS. In the Caucasian group, increased MPV (*p*-value: 0.046) had a risk-increasing effect, while increased WBC (*p*-value: 0.029) and neutrophil absolute count (*p*-value: 0.003) lowered PPMS risk. In the African American group, increased calcium (*p*-value: 0.028) and anion gap (*p*-value: 0.009) carried increased PPMS risk, and higher levels of chloride (*p*-value: 0.014) and bilirubin (*p*-value: 0.029) corresponded to decreased risk. Creatinine levels approached significance (*p*-values: 0.051 and 0.066) in both groups and carried increased risk. In the subtype analysis, all nominally significant tests for the Caucasian group were found in the CBC, and all nominally significant tests for the African American group were found in the CMP.

Multiple regression was performed for nominally significant lab values with closely related biology. In multiple regression analysis with the Caucasian mean platelet volume and platelet count as covariates, platelet count remained significant (*p*-value: 0.036), while mean platelet volume did not (*p*-value: 0.177). Similarly, in the African American group, in an analysis with anion gap, calcium, and chloride as covariates, anion gap remained significant (*p*-value: 0.032) while calcium (*p*-value: 0.331) and chloride (*p*-value: 0.153) did not. Additionally, in the same group, analyzing creatinine and bilirubin together resulted in bilirubin retaining its significance (*p* = 0.044); creatinine remained statistically insignificant (*p*-value: 0.083).

### 3.3. Random Forest Classification

ROC curves in Figure 1 summarize the predictive performance of the random forest model for each dataset. Both the CMP and CBC performed adequately at differentiating between MS cases and matched controls, with average AUC values of 0.725 and 0.710, respectively. However, when differentiating between PPMS and RRMS/SPMS patients, neither the CMP model nor CBC model achieved notably improved performance over the random guess baseline represented by the gray line, likely due to small sample sizes.

## 4. Discussion

In this study, we characterized differences in hematological profiles between MS cases and controls in both African American and Caucasian cohorts; furthermore, we performed a similar analysis to investigate differences between RRMS/SPMS and PPMS patients. A number of biomarkers differed between groups in our analyses, although we were unable to replicate the statistically significant relationships reported by Al-Hussain et al. Interestingly, increased MPV was not only associated with MS in both the Caucasian and African American groups, but it was also associated with PPMS in the Caucasian cohort in the subtype analysis and trended in the same direction for the African American group. Increased WBC had the opposite effect: in both groups of the case–control analysis and the Caucasian group in the subtype analysis, it carried a protective effect, and this directionality was also seen in the African American group during the subtype analysis, though not significantly. Possible explanations for the lack of significance in the African American subtype analyses include smaller sample sizes and physiological differences between the groups. In the African American analyses, higher anion gap was associated with both MS (case–control) and PPMS (subtype analysis). These results were not found in the Caucasian subtype analysis. To our knowledge, none of these relationships have been previously reported in the literature.

Previous studies have reported that bilirubin, creatinine, and the NLR have utility for distinguishing between MS cases and controls and between MS subtypes. Due to the MS subtypes present in our cohort, we were unable to compare our results to those previously reported by Ljubisavljevic et al. evaluating bilirubin as a biomarker of MS disease progression [21]. However, as both Ljubisavljevic et al. and Peng et al. reported, bilirubin levels were significantly reduced in MS patients in our dataset [10,21]. Despite the results of four previous studies that found that the NLR was elevated in MS patients, our case–control analysis found that higher NLR was associated with decreased risk of MS [24,25,26,27]. Two previous studies had reported inconsistent results regarding the ability of the NLR to predict disease course [23,25]. We found that the NLR was not associated with MS disease course. In our case–control analyses, creatinine was lower in MS patients, which contradicts the results in three other studies [22,28,30]. As with the NLR, conflicting evidence exists regarding the ability of creatinine to discriminate between MS subtypes [28,30]. While our results only approached significance, they indicated that creatinine was elevated in PPMS patients.

Given that the laboratory values in this study were selected based on availability rather than biological significance, it is encouraging that the case–control random forest models were able to perform notably better than the baseline. We expect that with sufficient samples improved performance from the subtype models would be observed, as well. Altogether, our results demonstrate that common laboratory values have utility in classifying MS cases and controls; larger samples are needed to assess their value in classifying patients based on MS subtype.

This study has several limitations. First, we were unable to account for the effects of medication use and other clinical characteristics (age of MS onset, disease duration) on hematological profile due to a lack of data for these variables. It has been reported that MS treatments can affect hematological values [31,32,33], so our observed associations may correlate with treatments and not necessarily disease onset. Observed associations may also be due to changes as MS progresses, rather than onset. Second, as we used EHR data, we were dependent on previously ordered tests, which limited our sample size for some of the less common tests. Our sample sizes were also limited by the relative rarity of MS in African American populations [1].

## 5. Conclusions

This study highlights several compelling trends; however, future studies are needed to replicate these findings while controlling for possible confounding factors and ensuring adequate study power for the subtype analyses.

## Figures and Tables

**Figure 1 ijerph-18-03318-f001:**
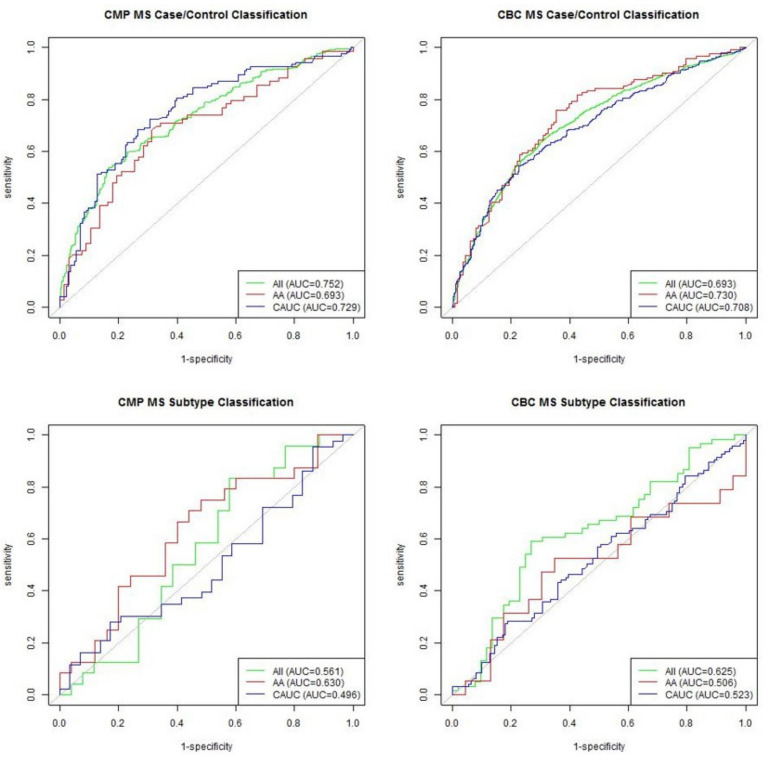
ROC curves showing performance of each random forest model.

**Table 1 ijerph-18-03318-t001:** Demographic and clinical data of the study populations.

		RRMS/SPMS	PPMS	All MS Cases	Controls
Caucasian Cohort	Number of patients	1643	259	4306	38,592
	Average Age (years)	47.1	53.6	56.3	58.2
	% Female	76.9%	68.0%	76.4%	51.9%
African American Cohort	Number of patients	227	47	596	3425
	Average Age (years)	44.0	52.4	52.5	51.1
	% Female	84.1%	68.1%	79.8%	61.6%

Age is calculated for the year 2018 using patient birth year.

**Table 2 ijerph-18-03318-t002:** Comparison of hematological profiles of MS cases and controls.

	Caucasian		African American	
	Odds Ratio (95% CI)	*p*-Value	Odds Ratio (95% CI)	*p*-Value
**CBC**				
Hemoglobin (Hgb)	**1.15 (1.13–1.18)**	**1.84 × 10^−39^**	**1.14 (1.08–1.21)**	**1.03 × 10^−5^**
Immature platelet fraction (IPF)	0.99 (0.95–1.03)	0.714	0.99 (0.88–1.08)	0.763
Immature reticulocyte fraction (IRF)	1.01 (0.99–1.03)	0.212	1.02 (0.98–1.07)	0.306
Mean corpuscular hemoglobin (MCH)	1.03 (1.01–1.04)	0.002	1.00 (0.97–1.03)	0.981
MHC concentration (MCHC)	1.04 (1.01–1.06)	0.011	0.91 (0.84–0.98)	0.010
Mean corpuscular volume (MCV)	1.01 (1.00–1.01)	0.041	1.01 (0.99–1.02)	0.396
Mean platelet volume (MPV)	**1.11 (1.07–1.16)**	**1.61 × 10^−6^**	**1.25 (1.12–1.40)**	**1.05 × 10^−4^**
Packed cell volume (PCV)	**1.06 (1.05–1.07)**	**2.80 × 10^−49^**	**1.06 (1.04–1.09)**	**9.10 × 10^−9^**
Platelet count (PltCt)	**1.00 (1.00–1.00)**	**7.46 × 10^−9^**	1.00 (1.00–1.00)	0.099
Red blood cell count (RBC)	**1.20 (1.12–1.27)**	**1.38 × 10^−8^**	**1.44 (1.23–1.68**)	**7.77 × 10^−6^**
Red cell distribution width (RDW)	**0.95 (0.93–0.98)**	**1.19 × 10^−4^**	0.92 (0.88–0.97)	0.003
RDW standard deviation (RDWSD)	1.01 (1.00–1.01)	0.191	1.00 (0.98–1.02)	0.880
Reticulocytes absolute value (RetAbs)	1.01 (1.00–1.02)	0.003	1.01 (0.99–1.03)	0.192
Reticulocyte Hgb equivalent (RETHE)	0.97 (0.94–1.01)	0.118	0.98 (0.91–1.04)	0.552
Reticulocyte count (RetiCt)	1.00 (0.92–1.08)	0.924	0.98 (0.71–1.23)	0.863
White blood cell count (WBC)	**0.95 (0.94–0.96)**	**4.55 × 10^−19^**	**0.94 (0.91–0.97)**	**1.67 × 10^−4^**
Neutrophil absolute count (NeutAbs)	**0.93 (0.91–0.94)**	**5.36 × 10^−27^**	‡	
Lymphocyte absolute count (LymAbs)	0.98 (0.95–1.00)	0.071	‡	
Neutrophil-to-lymphocyte ratio (NLR)	**0.95 (0.94–0.96)**	**4.93 × 10^−16^**	0.97 (0.93–1.00)	0.042
**CMP**				
Albumin (Alb)	**1.47 (1.34–1.61)**	**2.88 × 10^−16^**	**1.68 (1.30–2.18)**	**1.05 × 10^−4^**
Alkaline phosphatase (AlkP)	1.00 (1.00–1.00)	0.875	1.00 (1.00–1.00)	0.091
Anion gap (ANGAP)	**1.10 (1.08–1.12)**	**8.11 × 10^−37^**	**1.11 (1.07–1.16)**	**1.17 × 10^−6^**
Blood urea nitrogen (BUN)	1.01 (1.00–1.01)	0.010	1.00 (0.99–1.02)	0.555
Calcium (Ca)	1.05 (0.98–1.13)	0.148	1.17 (0.96–1.42)	0.115
Chloride (Cl)	**0.96 (0.95–0.97)**	**1.66 × 10^−13^**	0.98 (0.95–1.01)	0.255
Carbon dioxide (CO2)	1.01 (0.99–1.02)	0.375	1.00 (0.97–1.05)	0.826
Creatinine (Creat)	**0.48 (0.40–0.57)**	**2.32 × 10^−16^**	0.79 (0.64–0.92)	0.012
Glucose (Gluc)	**1.00 (1.00–1.00)**	**7.50 × 10^−5^**	**1.01 (1.00–1.01)**	**1.23 × 10^−4^**
Icterus index (IctIdx)	0.76 (0.34–1.39)	0.447	‡	
Potassium (K)	1.14 (1.04–1.26)	0.007	1.42 (1.10–1.83)	0.008
Lipid index (LipIdx)	1.01 (0.99–1.02)	0.465	1.02 (1.00–1.05)	0.079
Sodium (Na)	1.02 (1.01–1.04)	0.005	1.07 (1.03–1.12)	0.002
Aspartate amino transferase (SGOT)	1.00 (0.99–1.00)	1.03 × 10^−5^	1.00 (1.00–1.00)	0.471
Alanine amino transferase (SGPT)	1.00 (1.00–1.00)	0.444	1.00 (1.00–1.00)	0.947
Bilirubin (TBil)	0.19 (0.16–0.22)	3.7 × 10^−104^	**0.19 (0.13–0.29)**	**1.50 × 10^−14^**
Total protein (TProt)	1.11 (1.04–1.19)	0.002	0.99 (0.84–1.17)	0.920

After Bonferroni correction, the adjusted *p*-value for both groups was 0.0014. Significant tests are bolded above. Odds ratios represent risk of having MS. ‡ It was not possible to calculate the upper limit of the confidence interval.

**Table 3 ijerph-18-03318-t003:** Comparison of Hematological Profiles of RRMS/SPMS and PPMS Patients.

	Caucasian		African American	
	Odds Ratio (95% CI)	*p*-Value	Odds Ratio (95% CI)	*p*-Value
**CBC**				
Hemoglobin (Hgb)	1.03 (0.93–1.14)	0.579	1.08 (0.85–1.38)	0.520
Immature platelet fraction (IPF)	1.03 (0.90–1.38)	0.769	†	
Immature reticulocyte fraction (IRF)	1.05 (0.95–1.20)	0.433	†	
Mean corpuscular hemoglobin (MCH)	1.05 (0.97–1.12)	0.224	1.05 (0.92–1.20)	0.463
MHC concentration (MCHC)	1.14 (0.99–1.32)	0.070	0.99 (0.71–1.37)	0.947
Mean corpuscular volume (MCV)	1.01 (0.98–1.04)	0.547	1.03 (0.97–1.08)	0.344
Mean platelet volume (MPV)	**1.23 (1.01–1.51)**	**0.046**	1.20 (0.76–1.94)	0.442
Packed cell volume (PCV)	1.00 (0.97–1.04)	0.837	1.03 (0.95–1.13)	0.454
Platelet count (PltCt)	**1.00 (1.00–1.00)**	**0.022**	1.00 (0.99–1.00)	0.435
Red blood cell count (RBC)	1.02 (0.74–1.39)	0.926	0.92 (0.47–1.83)	0.808
Red cell distribution width (RDW)	0.91 (0.82–1.03)	0.129	0.99 (0.80–1.27)	0.959
RDW standard deviation (RDWSD)	0.99 (0.95–1.03)	0.456	1.00 (0.91–1.10)	0.930
Reticulocytes absolute value (RetAbs)	‡		†	
Reticulocyte Hgb equivalent (RETHE)	0.98 (0.78–1.18)	0.825	†	
Reticulocyte count (RetiCt)	0.80 (0.51–1.33)	0.358	0.98 (0.19–5.29)	0.983
White blood cell count (WBC)	**0.95 (0.90–0.99)**	**0.029**	0.92 (0.81–1.06)	0.254
Neutrophil absolute count (NeutAbs)	**0.91 (0.86–0.97)**	**0.003**	0.86 (0.72–1.03)	0.092
Lymphocyte absolute count (LymAbs)	0.99 (0.83–1.19)	0.945	0.69 (0.45–1.07)	0.090
Neutrophil-to-lymphocyte ratio (NLR)	1.00 (0.96–1.06)	0.984	0.93 (0.77–1.17)	0.497
**CMP**				
Albumin (Alb)	1.02 (0.68–1.50)	0.927	1.88 (0.84–4.17)	0.112
Alkaline phosphatase (AlkP)	1.00 (0.99–1.00)	0.738	0.99 (0.98–1.00)	0.122
Anion gap (ANGAP)	1.05 (0.97–1.14)	0.220	**1.40 (1.09–1.82)**	**0.009**
Blood urea nitrogen (BUN)	1.01 (0.98–1.04)	0.683	1.03 (0.96–1.11)	0.487
Calcium (Ca)	1.09 (0.81–1.45)	0.553	**2.58 (1.12–6.16)**	**0.028**
Chloride (Cl)	1.03 (0.98–1.08)	0.249	**0.82 (0.70–0.96)**	**0.014**
Carbon dioxide (CO2)	0.95 (0.90–1.01)	0.123	1.08 (0.90–1.30)	0.396
Creatinine (Creat)	2.23 (1.05–5.13)	0.051	4.78 (1.04–29.11)	0.066
Glucose (Gluc)	1.00 (1.00–1.01)	0.714	1.01 (1.00–1.03)	0.110
Icterus index (IctIdx)	0.75 (0.19–2.63)	0.656	†	
Potassium (K)	1.21 (0.80–1.83)	0.376	1.03 (0.31–3.50)	0.965
Lipid index (LipIdx)	1.07 (0.99–1.22)	0.226	†	
Sodium (Na)	1.02 (0.96–1.09)	0.461	0.96 (0.79–1.16)	0.692
Aspartate amino transferase (SGOT)	1.01 (1.00–1.02)	0.309	1.02 (1.00–1.08)	0.422
Alanine amino transferase (SGPT)	1.00 (1.00–1.01)	0.353	1.03 (0.99–1.07)	0.191
Bilirubin (TBil)	0.88 (0.67–1.41)	0.443	**0.22 (0.05–0.83)**	**0.029**
Total protein (TProt)	1.06 (0.76–1.46)	0.730	1.79 (0.93–3.44)	0.076

After Bonferroni correction, the adjusted *p*-value was 0.0014 for the Caucasian group and 0.0017 for the African American group. No tests were significant with these *p*-values. Nominally significant tests are bolded above. Odds ratios represent risk of having PPMS compared to RRMS/SPMS. † Too few patients had data available to perform the analysis. ‡ It was not possible to calculate the upper limit of the confidence interval.

## Data Availability

Data is part of the Vanderbilt University Synthetic Derivative and can be accessed via approval by Vanderbilt University.

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
