# Peer review of "The Impact of Multiple Sclerosis Disease Status and Subtype on Hematological Profile"

_ijerph, 2021, doi:10.3390/ijerph18063318_

Round 1

Reviewer 1 Report

his study analyses the association between Complete Blood Count (CBC), Comprehensive Metabolic Panel (CMP) tests and Multiple Sclerosis occurrence/subtype classification by evaluating large data sets obtained from MS and control cohorts grouped according to their belonging to Caucasian or African-American group. The authors refer the presence of some altered values in the MS group, while no significant correlation with MS clinical subtypes has been observed. These observations do not confirm previous reports obtained in small cohorts as in Al Hussain paper.

The development of simple and valuable criteria to better identify/classify MS individuals is lacking and expected to significantly improve clinical MS management.  Moreover, the possibility to identify by random forest classifier approach (largely employed in various clinical contexts) new clinical biomarkers using commonly employed hematological panels is an intriguing aim to be performed.

A major concern related to this analysis strategy is the impossibility to specifically relate alteration of commonly employed hematological parameters (i.e. creatinine, bilirubin or blood count) per se to MS occurrence; more useful might be the investigation on the possibility that biochemical alterations might indicate MS progression along clinical follow up of the disease. These issues need to be at least more deeply discussed in the paper.   

Author Response

Thank you for your comments, especially regarding the stated relationship between associations and MS occurrence. We concur that with our dataset, we are unable to attribute the changes in patients with MS compared to controls to onset, and it could be attributable to disease progression or treatments. We have included this in our discussion.

Reviewer 2 Report

Jacob M. Miller et al. produced a very well-written article focused on “The Impact of Multiple Sclerosis Disease Status and Subtype on Hematological Profile”. I consider the manuscript very fascinating but, in the same time, I suggest several revisions needed to improve the reliability and the completeness of the paper:

In “Introduction” section, the authors said: “Random forest classifiers have been used to discover novel disease-biomarker relationships and to classify patients in a variety of settings”. I suggest the authors to add more recent literature regarding the application of random forest algorithms to biomarker discovery, such as non-coding RNAs and mitochondrial transcripts, and to biochemical in-silico pathway analyses. Regarding this, I suggest to add the following papers PMID: 33233726 and PMID: 33233546.

Moreover, in order to improve the innovation of statistical approach, I suggest to briefly describe the limits of already published case-study works that exploited only traditional statistical methods. Regarding this, I suggest the authors to cite interesting bibliography like PMID: 27737651 

Finally, manuscript requires English revisions and typos correction.

Author Response

Thank you for your comments and the direction to further literature. We have incorporated this into our Introduction.

Reviewer 3 Report

The purpose of this paper was to evaluate if patient hematological profiles (blood counts and metabolic values) were good biomarkers to identify differences between MS cases and controls and between patients with different MS subtype (RR vs PP).

Additionally, random forest models were used to determine the aggregate utility of common hematological tests in determining MS disease status and subtype. The authors achieved some significant results only when comparing MS cases and controls.

Comments:

Data were de-identified records from a single Medical center, increasing homogeneity of lab data.

The sample size seems ok, at least for the comparison MS vs controls.

The lack of information about age and other medical characteristics is a severe limit, as the authors pointed out.

Statistical methods

Logistic regression models are appropriate as well as the Bonferroni correction applied at group level and details are present in the methods and results sections.

What is not very clear to the reviewer is the multiple regression analysis. It is mentioned in the results but not in the methods section (at least this is what I had found). And there are no tables presented in the manuscript or in the supplemental material. Some correlation between biomarkers is expected, as the authors pointed out, and it need to be described using more details.

The machine learning part (random forest model) could be interesting but it lacks of details. Which are the standard parameters used for example?

A ROC curve AUC close to 70% usually does not mean good prediction as far as I know, it is just sufficient.

Author Response

Thank you for your comments on the methods and results, especially regarding the missing methods for the multiple regression analyses. We have included those details. Only two analyses were run with multiple regression and the results and included in the text only. We have also updated the language discussing the AUC to be sufficient, instead of good.

Reviewer 4 Report

This is a well written and presented paper. However, the study has multiple flaws that prohibit from drawing scientific sound conclusions. The absence of data relative to the treatment, treatment duration, disease duration, EDSS,… are all critical factors needed to draw conclusions comparing Control to MS patients as well as the different subtypes of MS disease. For example, stating that “increase MPV correspond to an increase risk of MS” is an erroneous conclusion based on the data presented in this manuscript. IFN-b treatment, the most widely used treatment for MS patients, is known to reduce the level of platelets (Koudriavtseva T et al., 2015). Other studies have also shown the effect of MS treatments on WBC (Rieckmann P et al., 2004, Lim ZW et al., 2016,…). The authors aimed to draw conclusions based on presence or absence of the disease, but it is absolutely wrong to ignore the other factors that would impact CBC and CMP analysis. 

Author Response

Thank you for your comments regarding the conclusions drawn from the study. We agree completely and have updated the wording in the discussion to reflect the potential for treatments or MS disease progression to be responsible for the associations observed, including references. 

Round 2

Reviewer 2 Report

The authors addressed all suggested points. Well done.

Reviewer 4 Report

There is very little changes between this new submission and the previous one. My comments are still the same: the absence of data relative to the treatment, treatment duration, disease duration, EDSS,…are all critical factors needed to draw conclusions comparing Control to MS patients as well as the different subtypes of MS disease. The authors aimed to draw conclusions based on presence or absence of the disease, but it is wrong to ignore the other factors that would impact CBC and CMP analysis. 

Author Response

We agree with reviewer 4 that there are many other factors that could influence laboratory values. Unfortunately, these data were beyond the scope of this analysis and dataset. Comments in the discussion should better reflect this.